# A Preliminary Study to Classify Corn Silage for High or Low Mycotoxin Contamination by Using near Infrared Spectroscopy

**DOI:** 10.3390/toxins14050323

**Published:** 2022-05-03

**Authors:** Francesca Ghilardelli, Mario Barbato, Antonio Gallo

**Affiliations:** Department of Animal Science, Food and Nutrition (DIANA), Faculty of Agricultural, Food and Environmental Sciences, Università Cattolica del Sacro Cuore, 29122 Piacenza, Italy; francesca.ghilardelli@unicatt.it (F.G.); mario.barbato@unicatt.it (M.B.)

**Keywords:** random forest, machine learning, emerging mycotoxins, forage

## Abstract

Mycotoxins should be monitored in order to properly evaluate corn silage safety quality. In the present study, corn silage samples (n = 115) were collected in a survey, characterized for concentrations of mycotoxins, and scanned by a NIR spectrometer. Random Forest classification models for NIR calibration were developed by applying different cut-offs to classify samples for concentration (i.e., μg/kg dry matter) or count (i.e., n) of (i) total detectable mycotoxins; (ii) regulated and emerging *Fusarium* toxins; (iii) emerging *Fusarium* toxins; (iv) *Fumonisins* and their metabolites; and (v) *Penicillium* toxins. An over- and under-sampling re-balancing technique was applied and performed 100 times. The best predictive model for total sum and count (i.e., accuracy mean ± standard deviation) was obtained by applying cut-offs of 10,000 µg/kg DM (i.e., 96.0 ± 2.7%) or 34 (i.e., 97.1 ± 1.8%), respectively. Regulated and emerging *Fusarium* mycotoxins achieved accuracies slightly less than 90%. For the *Penicillium* mycotoxin contamination category, an accuracy of 95.1 ± 2.8% was obtained by using a cut-off limit of 350 µg/kg DM as a total sum or 98.6 ± 1.3% for a cut-off limit of five as mycotoxin count. In conclusion, this work was a preliminary study to discriminate corn silage for high or low mycotoxin contamination by using NIR spectroscopy.

## 1. Introduction

Silage is produced to preserve forage with high moisture content by controlled fermentation and represents an important feed for cattle. Corn (*Zea mays* L.) is the most widely grown crop for silage [1,2] mainly due to its high-yielding, consistent source of forage, high digestibility, good animal palatability, and ability to be successfully ensiled. As a result, most dairy farms in northern Italy have adopted a cropping plan mainly based on the cultivation of corn hybrids for whole-crop plant production, thus increasing the self-sufficiency of the energy requirements [2] 

The microbiota in ensiled mass changes during conservation due to modification of temperature, humidity, oxygen availability, and pH conditions that normally occur in different ensiling stages. Further, poor storage conditions can also lead to undesirable mound contamination, mycotoxin production, and a reduction in nutritional value [3,4].

In particular, mycotoxins are secondary metabolites produced by fungi mainly belonging to *Aspergillus*, *Fusarium*, *Alternaria*, and *Penicillium* strains [5,6]. Corn silage can be co-contaminated by several mycotoxins with negative consequences on animals, such as a reduction in feed intake and milk yield and quality or reproductive problems [7]. Importantly, ruminants are less affected by mycotoxins than monogastric animals thanks to their ability to degrade mycotoxin mother molecules [8,9]. Rumen microflora and feed particles contained in the rumen-reticulum compartment are capable of degrading, deactivating, or binding toxic molecules. In particular, a study on monogastric animals showed that mycotoxins can compromise several intestinal functions, such as digestion, absorption, and permeability, and can result in lower productivity and poor health [10]. Due to the increasing use of silage in monogastric diets (e.g., swine or avian species such as broilers, hens, geese, etc. [11]), its safety represents a pre-requisite for silage use in animal diets, not only for ruminants but for other species as well.

Therefore, animal diets include starch and protein feeds, their by-products, and, mainly for ruminants, grazed forage, hay or grass, grass/legume, whole-crop forage maize, high moisture ear corn, high moisture corn, and small grain and sorghum silages [12]. Mycotoxins could contaminate all these feeds and the use of roughages in ruminants can increase the risk of mycotoxin exposure compared to swine and poultry that usually have less varied diets. Some evidence [13,14] reported the main exposure of cattle to mycotoxins is usually related to the consumption of contaminated silage, even if this aspect remains poorly investigated. To overcome this safety problem, inoculation of silage with lactic acid bacteria (LAB) before ensiling is a common practice to improve the fermentation quality and increase its aerobic stability [15,16,17]. However, the antifungal effects of this practice are uncertain, particularly in the oxygen-rich microenvironment of silage in which caked and clumpy areas develop visible green-gray mold indicative of mycotoxin contaminations, such as aflatoxins and several *Fusarium*-produced mycotoxins [18,19,20].

Despite the potential benefits of using LAB inoculants for preserving silage safety, Ogunade et al. [4] reported that few studies have examined the use of mold-inhibiting chemical additives or microbial inoculants to prevent or reduce mycotoxin contamination during different ensiling stages. As recently reported [21,22], some LAB were able to reduce aflatoxin contamination in corn silage, whereas the same tested strains did not reduce these mycotoxins in high moisture corn. Furthermore, analysis of levels of mycotoxins detectable in silage indicated that LAB inoculants interacted with several fungal populations by changing the mycotoxin profile relative to untreated silage, thereby increasing levels of some mycotoxins and decreasing levels of others [16,17,20]. Consequently, future investigations should examine the relationship between LAB and mycotoxigenic fungi during ensiling in an effort to develop methods to produce safe silage.

Hence, global mycotoxin regulations are essential to keep levels of mycotoxin low while following the recommended agricultural, storage, and processing practices. However, regulators have focused on a few groups of mycotoxins while several “emerging toxins” can contaminate both concentrate and forage [23]. Further, feeds are rarely contaminated by only one mycotoxin, but more often by several mycotoxins simultaneously. A recent survey carried out on corn silage showed multi-mycotoxin contamination and indicated that 47% of samples contained five or more mycotoxins out of the 22 analyzed [24]. 

The quantification of both regulated and emerging toxins is necessary to evaluate feed safety. The most common approach to monitoring mycotoxins relies on chromatographic and immunological methods [25,26]. Such techniques allow for accurately quantifying multiple mycotoxins. However, these approaches are expensive and time-consuming due to the extraction and clean-up steps. Alternative cheaper and faster methods have been proposed and are increasingly being applied to complement the standard analytical technique [27]. Emerging mycotoxin detection/sorting techniques include biosensors for cereal [28], nano-biosensors for feed [29], -omics tools [30], and other rapid and non-destructive quantifications (see Fumagalli et al. 2021 for a review).

Recently, Mota et al. [31] reviewed the use of sensor-based electronic nose systems for mycological analysis. The review shows that sensor-based electronic nose systems are mostly focused on the food industry and targeting the genus *Penicillium* in particular, but contaminations from *Aspergillus* and *Fusarium* are also investigated via the sensor-based electronic nose. Near-infrared spectroscopy (NIR) and an electronic tongue and electronic nose have been at the forefront of quality control technologies in the food industry due to their real-time data processing capabilities [32]. As an example, the use of an electronic nose was recently applied for assessing aflatoxin B1 and fumonisins contamination in 316 and 229 corn samples, respectively, achieving the best classification result with the artificial neural network. The aim was to investigate whether the electronic nose was capable of grouping samples contaminated at levels above or below the legal limits, achieving correct estimation for aflatoxin B1 in about 77% of samples and 78% of correct classifications for Fumonisins [33].

The first application of infrared spectroscopy to the analysis of microorganisms can be traced back to the 1950s [34,35]. Moreover, in the 1980s, papers on the use of NIR to detect fungal contamination [36] or for quantifying mycotoxins in barley, corn, and wheat [37] were published. More recently, a linear discriminant analysis was applied to determine Aspergillus spp. contamination levels in peanuts by comparing NIR spectroscopy and an electronic nose [38]. Other studies conducted on brown rice samples artificially inoculated with *Aspergillus flavus* and *A. parasiticus* strains of fungus, demonstrated that NIR and MIR technology had the potential to simultaneously detect aflatoxin B1, B2, G1, and G2 while achieving good predictive accuracy for both NIR (R^2^ = 0.936–0.973, RPD = 2.5–4.0) and MIR spectroscopy (R^2^ = 0.922–0.970, RPD = 2.5–4.0) [39]. Therefore, Infrared (IR) spectroscopy can be used to quickly assess the hygiene quality of the feeds, other than their chemical composition. In particular, the IR spectroscopy is rapid, non-destructive, versatile, often applicable on-field, and requires no chemicals. All these characteristics have led to its use in multiple applications [40]. 

As reported by Min and Cho [41], spectroscopic techniques have proven to be alternative tools for the early detection of mycotoxins in agricultural products, and their advantages over conventional invasive methods are related to their rapidity and non-destructive characteristics. Because of the deleterious health effects of mycotoxins, the quick monitoring of animal feed contamination can avoid economic losses while preserving animal health. The use of this technology favors quick feedback in the field, and a larger and more frequent number of analyses may be performed, ensuring greater safety in the use of the cereal in both animal and human nutrition [42]. Moreover, the use of so-called “rapid methods” is highly relevant for improving knowledge on the presence and distribution of mycotoxins in food and feed. NIR analysis represents one of the most promising tools for farm screening silage mycotoxin tools as affirmed by Cheli et al. [18]. 

Quantification models can lack the sensitivity necessary to determine mycotoxin contamination, due to the small size of mycotoxins molecules [25] present in low concentrations in the range of ppm (mg/mL) or a few parts per billion (ppb; ng/mL), and infrared spectroscopy currently is not always sufficiently sensitive for this quantitative purpose [37]. 

According to previous reports describing the strengths and criticalities of the application of NIR spectroscopy in the quantification or discrimination of mycotoxins, an alternative approach was proposed to develop qualitative discriminant models for the classification of mycotoxins [19,20]. Due to the difficulty of the NIR spectrometer to develop robust regression models for the quantification of individual mycotoxins, and in relation to the fact that silages are usually subject to co-contamination of these toxins, the aim of the current study was therefore to obtain qualitative models able to discriminate silage for total concentrations or total counts of groups of mycotoxins. In order to rank silage based on mycotoxins co-contamination, we calibrate a NIR apparatus using the Random Forest (RF), a machine learning classification technique.

## 2. Results

The acquired spectra were assessed using Principal Component Analysis (PCA) to identify putative outliers. No samples had values both outside the F-residuals and Hotelling’s T2 threshold values. Consequently, no spectrum was removed (Figure 1a).

The samples included in calibration were characterized for chemical, fermentative, and organoleptic traits and mycotoxin contamination. Chemical and fermentative traits are reported in Table 1 and data were adapted by Gallo et al. [20]. 

The variables included in the NIR calibration were the count and sum (i.e., concentration) of different secondary metabolites produced by some fungal organisms. The NIR calibration concerned total counts of mycotoxins and the sum of single mycotoxin concentrations of mycotoxigenic fungi species such as *Aspergillus*, *Fusarium*, *Penicillium*, and *Alternaria*. Other variables included in calibration were regulated and emergent *Fusarium*-produced mycotoxins, only emerging *Fusarium*-produced mycotoxins, Fumonisins mycotoxins produced by fungi of the *Fusarium* genus, and lastly, *Penicillium* produced mycotoxins.

The contamination levels of the original 115 corn samples are reported in Table 2. 

The samples included in this study rank between low-l to high-contamination levels from a minimum of 14 to a maximum of 45 types of mycotoxins detected in the 115 corn silages, with a mean of 26.20 ± 6.42 and a 25% and 75% quartile of 21.50 and 30.50, respectively. Moreover, the average concentration of total mycotoxins was 5895.70 ± 7252.46 μg/kg DM. The mean concentration of emerging *Fusarium* toxins was 2453.83 ± 3571.47 μg/kg DM. Conversely, the mean concentration of regulated and emerging *Fusarium* toxins was 4781.04 ± 6539.44 μg/kg DM, whereas the counts of this mycotoxin category were 15.33 ± 3.61. Concerning all fumonisins mycotoxins, the range of concentration values was 16,104.84 μg/kg DM for a maximum of eight co-occurrence of these, and high fumonisins occurrences were observed for non-regulated FB3, FB4, FA1, and masked forms of FA1. *Penicillium* toxins ranked between a minimum of 2.12 to a maximum of 1286.49 μg/kg DM. Ochratoxin A was not detected in any samples, and mycophenolic acid and/or its metabolite mycophenolic acid IV was found in only five samples.

### Calibration Model Results

In Table 3, we reported the performances of the total sum and mycotoxin count calibrations for each cut-off value for the balanced validation dataset. The average accuracy ranked from 82.2 ± 5.9%, using a cut-off of 4000 μg/kg DM, to 96.0 ± 2.7%, adopting the highest cut-off value of 10,000 μg/kg DM. Sensitivity and specificity were >80%, independently, if the threshold applied to discriminate the two classes was 4000, 7000, or 10,000 μg/kg DM. The lowest sensitivity and specificity (80.2 ± 8.5% and 81.3 ± 8.0%, respectively) were obtained using 4000 μg/kg DM as a threshold, whereas the best model performance was achieved using 10,000 μg/kg DM as a threshold.

In regard to the total count of mycotoxins in corn silage samples, the best model was achieved using 34 co-occurring mycotoxins to discriminate the high vs. low classes. In this model, misclassification of the upper contamination was negligible with 0.03 ± 0.30 samples wrongly attributed to class 1 and 2.7 ± 1.7 to class 2.

In Table 4, we report all the calibration models related to regulated or emerging *Fusarium*-produced mycotoxin (i.e., R&E-*Fusarium* toxins). Inferior performances were achieved using cut-off limits of 1500 μg/kg DM and 13 for sum or count, respectively. Accuracy levels were lower than 80% in these two models, although we recorded an improvement in ranking samples within the class with contamination levels of 2500 μg/kg DM for the *Fusarium*-mycotoxin contamination sum and 15 for the *Fusarium*-mycotoxin count. In both cases, the sum and count NIR calibration showed a relatively narrow confidence interval (95%) of about 77.9 ± 4.7%, 93.2 ± 2.9% and 71.1 ± 5.6%, 90.1 ± 3.8%, respectively.

Regarding mycotoxins that have no specific regulations (Table 5), satisfactory calibration models for sum parameters were developed using the three different thresholds of 700, 1000, and 1200 μg/kg DM; however, in the latter case, the accuracy achieved was ~90% with a 95% CI of 83.2 ± 3.8%, 95.4 ± 2.2% and a precision of 94.9%. Similarly, calibration for a count of emerging *Fusarium*-mycotoxin showed good performances when the limit of class determination was eight co-occurring emerging mycotoxins. In this case, only 3.9 ± 2.2 low-contamination samples were erroneously attributed to the high-contamination category.

Concerning the fumonisins sum NIR calibration (Table 6), the average level of accuracy was 88.3 ± 4.3% with a precision average of 92.4%. In the same classification, the allocation by the model to class 2 achieved, on average, a precision of 84.6%.

Regarding the same mycotoxin contamination parameter, setting 6 as the classification limit allowed us to obtain a good classification RF model. The accuracy was 92.8 ± 3.3%; a few samples (i.e., 1.0 ± 1.4) were misclassified in class, with low co-occurrence contamination, and the specificity for the high level was 97.6 ± 3.1%.

The last calibration was about *Penicillium*-produced mycotoxin concentration with the count reported in Table 7. The accuracy values of both threshold contaminations of 250 and 350 μg/kg DM ranged from 90 to 95%, with an average of 92.9% and 95.1%, respectively. Better results were obtained both for sensitivity and specificity using 350 μg/kg DM concentration as the class limit (i.e., 91.0 ± 4.9% and 99.7 ± 1.6%, respectively).

Adopting three or five as a cut-off in terms of count class division increased the accuracy by 18.3% with no misclassified samples in class 1.

## 3. Discussion

The assessment of multiple mycotoxin contamination in feedstuffs and total mixed rations should be considered when formulating dairy cow diets, albeit those mycotoxins may be below regulatory limits [43]. In the current manuscript, we developed different NIR calibrations to provide a tool to rapidly assess mycotoxin co-contamination risk to animals, with a special focus on corn silage total contamination. The mean values of nutritional composition or fermentative traits were typical for corn silage produced in this geographical area [20]. The mycotoxins included in the calibration belong to the main mycotoxigenic fungi *Aspergillus* spp., *Fusarium* spp., and *Penicillium* spp. which are prevalent in corn silage [44]. In the current manuscript, we aggregated mycotoxins produced by the same fungal genera and quantitatively measured them using the LC-MS/MS reference method. In order to have more information regarding the specific mycotoxins that were detected in our samples, we referred Gallo et al. [20] for details. The proposed calibration aims to develop a rapid and non-destructive method to classify corn silage into low or high contamination classes, representing a qualitative clustering technique.

### 3.1. Mycotoxin’s Contamination Occurrences and the NIR Calibration Approach

In particular, *Aspergillus* produced the mycotoxins rugulusovin or brevianamide F, as detected in most of the corn silages, with an incidence >88.9%. Despite these compounds being assigned to this class, they represent the unspecific diketopiperazine produced by many microbial species among several *Aspergillus* strains. No aflatoxins or nigragillin were detected, but emodin was detected at low concentrations in more than 50.0% of samples. Kojic acid was found in most of the corn silages, and incidences lower than 21% were calculated for 3-nitropropionic acid, averufin, bis(methylthio)gliotoxin, and asperphenamate [20]. *Fusarium* produced mycotoxins were quantified and the regulated FB1 and FB2 were detected in the majority of the corn silages with occurrence values higher than 95.5%; high fumonisin occurrences were also observed for non-regulated FB3, FB4, FA1, and masked forms of FA1, their incidences being higher than 54.2%. The average concentrations of FB6 and FA2 results were lower than 12.4 μg/kg DM, and ZEA was detected in all samples with incidences ranging from 22.2% to 100.0%. No trichothecenes type-A, regulated T-2, and HT-2 toxins were detected; trichothecenes type-B and DON were detected as well as nivalenol and deoxynivalenol-3-glucoside, but at low concentrations and low incidences. Other than regulated *Fusarium* produced mycotoxins, siccanol, moniliformin, equisetin, epiequisetin, and bikaverin were detected in the majority of analyzed corn silages. Finally, enniatins were detected in some samples, while Beauvericin was quantified in the majority of corn silages with an incidence of >83.3%. Regarding *Penicillium* produced mycotoxins, ochratoxin A and barceloneic acid were not detected in any samples, whereas 7-hydroxypestalonic, pestalotin, oxaline, phenopyrrozin, and questiomycin A were detected at low concentrations (<50.0 μg/kg DM) in several corn silages, with incidences higher than 60%. Mycophenolic acid, as well as its metabolite mycophenolic acid IV, were detected only in two samples. 

Since corn silage could be contaminated by a high number of regulated and emerging mycotoxins, it was decided to calibrate the NIR spectrometer in relation to their concentration or count. The samples included in the study were randomly collected from livestock production realities representative of the intensive dairy farm system in Italy and consequently, their distribution in class was biased towards low levels of contamination. One of the major obstacles in mycotoxin modeling is related to mycotoxin monitoring data often being unbalanced towards low mycotoxin concentrations [45]. To obtain more robust and accurate calibrations, a re-balancing approach was adopted to obtain class-balanced distribution. Re-balancing has been previously adopted in NIR calibration using different mathematical approaches, but ultimately by under-sampling the majority class and/or generating a new representative of the minority class [46,47,48]. Moreover, a recent study evidenced that the combination of oversampling and downsampling techniques performed better than using exclusively one or the other [49].

Here, we applied the “smote and undersample” algorithm to oversample the minority class by SMOTE while undersampling the majority class, as this method was specifically conceived to handle extremely imbalanced data [50]. The RF classification model was then applied as it has been successfully applied by others to classify single corn kernels with aflatoxin contamination from NIR spectra [51]. To evaluate real model performances not related to the possible random good distribution of samples, the process was repeated 100 times and the results showed were the mean of these repetitions. 

### 3.2. NIR Calibrations

The model developed in this study allowed us to obtain a good calibration in terms of accuracy, sensitivity, and specificity thanks to its ability to balance errors in datasets where classes are not equally distributed [52,53]. Such good performances were obtained thanks to the correlation between the NIR spectra for various functional groups and major fungal constituents. 

*Fusarium* mycotoxins are primarily produced by *F. proliferatum* and *F. verticillioides* and FB1 and FB2 appeared to be the most predominant among fumonisins [54]. FB1 and FB2 were detected in almost all samples included in this calibration; FB6 and FBA2, on the contrary, showed low incidences in our data. The ability of the NIR spectrometer to see *Fusarium* mycotoxins was supported by previous relatively good qualitative prediction using NIR spectrometer obtained for *Fusarium verticillioides* fungal infection, ergosterol, and Fumonisin B1 content with model performances from R^2^ = 0.78 for *F. verticillioides* and Fumonisin B1 to R^2^ = 0.81 for ergosterol content [53].

*Fumonisin* contamination in corn silage is usually ascribed to pre-harvest crop conditions [4,18], but management strategies before harvest, can increase *Fumonisin* contaminations. Here we see the importance of discriminating corn silage based on total *Fumonisin* mycotoxin contamination. This kind of forage is often infected by mycotoxin-producing *Fusarium* fungi, first among all DON and ZEN but also their modified forms. In fact, positive correlations were established between concentrations of the co-occurring mycotoxins and their modified forms, and to avoid underestimation, it is necessary to also quantify modified mycotoxins; more than half of all forage corn samples (i.e., 57%) included in the study were co-contaminated with DON, ZEA, and other modified forms because DON and ZEN are produced by the same Fusarium species (*F. graminearum* and *F. culmorum*) [55]. Moreover, the great accuracy in predicting total contamination of mycotoxin in terms of count or concentrations was supported by comparative studies using NIRS and imaging methods and concluding that NIR has good recognition of heavily mold-infected and lower infected kernels [56]. Further, the proliferation level of these toxic compounds in forage was strictly related to milk in its chemical composition with an accumulation of sphingolipids, together with purine and pyrimidine derivatives, in bad quality corn samples [57], which supports the importance of quickly determining mycotoxin corn silage contamination.

The performances achieved for fumonisin mycotoxin contamination were in line with those reported by Levasseur-Garcia et al. [58] on 117 corn samples collected in Italy, Denmark, France, Hungary, The Netherlands, and Poland with a percentage of well-classified samples of 96%. Similar predictive capacities of NIR towards contamination by both fumonisins and ZEA were found by Tyska et al. [59]. In this study, a total of 676 Brazilian corn samples were employed to calibrate the NIR using 236 samples for FBs and 440 for ZEN contamination, achieving an R^2^ of 0.899 and 0.984, respectively. 

The complex mixture of mycotoxins in silages can originate from pre-harvest mold contamination, in particular for *Fusarium* spp., as well as from post-harvest contamination with toxins produced by fungal species such as *Aspergillus* and *Penicillium* [4]. Because of *Penicillium*’s ability to grow in silage conditions (i.e., low oxygen and high carbon dioxide concentrations, low temperatures, and high concentrations of volatile fatty acids), the determination of these fungi mycotoxins is particularly important. Importantly, *P. roqueforti* is the most frequently occurring toxicogenic fungal species in Europe [60]. Despite the concentration in samples included in the present study being lower than levels associated with animal disorders, the classification risk of *Penicillium* toxins is important to evaluate the co-contamination in corn silage. 

In all calibrations, the ability of NIR to differentiate between low contamination and high contamination toxins was greater using higher threshold classes both for the count and for the sum. NIR spectroscopy was able to provide information about the chemical functional groups in the molecules [37]. A fungal attack damages tissues and cells so the changes in corn silage properties, such as protein, carbohydrates, and lipid content, were related to changes in fungal contamination as reflected in their spectral signature [61]. Adopting a greater level of cut-off allows for better sorting between low and high contaminated corn silages. 

Critically, the current NIR calibrations were proposed as practical tools useful to classify corn silage by their contamination by developing qualitative predictive models. Both mycotoxins and fungi cannot be directly detected by NIR spectroscopy due to the lack of sensitivity of this method, but indirect information regarding the contaminations could be noted, e.g., changes related to alterations due to fungal attack are visible using the NIR system [62]. Another strength of this method is the possibility of performing simultaneous analysis of the different types of mycotoxins arising from the contamination of different fungi. To improve the proposed method, and make it more robust and applicable, it is necessary to expand the initial database of 115 samples. Additionally, the use of Vis/NIR spectroscopy can further improve the calibrations [63]. Functional groups (CH, NH, and OH) in chemical components could exhibit characteristic absorption in the Vis/NIR region which would result in significant changes in specific spectral bands during fungal growth. These changes could be reflected by Vis/NIR spectrum and NIR hyperspectral imaging (HSI) as reported by Mishra et al. [64].

### 3.3. Practical Application

In order to better explain the potentialities of our method, we report some recent studies published in the last three years showing the great practical-applicative potentialities of the use of NIR in monitoring mycotoxin contamination in several matrices used in animal or human nutrition in Table 8. In particular, the results obtained in our study were in line with the classification studies reported below [65,66,67,68,69,70,71]. The study reported here gave a practical application of NIR calibration in order to underline the importance of this type of approach as a practical tool to assess the safety of silage. For a deep overview of the previous studies using NIR for the prediction of mycotoxin in different feeds and food, please refer to Levasseur-Garcia [37].

## 4. Conclusions

NIR spectroscopy has several advantages with respect to traditional analytical methods, mainly due to its capability of being a rapid, non-destructive, and economical tool capable of estimating several chemical-biological parameters simultaneously. Its application to classify or estimate fungal and mycotoxin contamination has received more and more attention, however, the very low concentration levels of mycotoxin could limit the effective detection and quantification of corn silage contamination. The NIR method developed in the current paper was based on the detection of the sums or counts of a different group of mycotoxins, while not focusing on a specific fungi toxin. Coupled with the contamination class re-balancing step, this method produced good calibration models in terms of accuracy, sensitivity, and specificity and appears to be a suitable screening method to provide rapid information regarding silage mycotoxin contamination. Samples belonging to a high contaminated class should be successively analyzed by conventional methods in order to assess the real risk of mycotoxin contamination for regulated animals in particular. 

## 5. Materials and Methods

### 5.1. Sample Collection, Preparation, and Analysis

A total of 115 corn silage samples were collected in a survey from dairy farms located in the Po Valley (Italy) and Sardinia. These farms were randomly selected and visited over the years 2017–2019. The collection was conducted by adopting a sampling procedure based on methods described by Undersander et al. 2005 from horizontal bunker silos. The wet sample was about 2 kg and was sampled from at least four different points of each bunker feed out face, placed in a clean plastic jar, hermetically closed, and analyzed as reported below.

Each sample was dried at 60 °C in a ventilated oven until achieving a constant weight, then milled through a 1-mm and 0.5-mm screen using a laboratory mill (Thomas-Wiley, Arthur H. Thomas Co., Philadelphia, PA, USA), and stored for analysis. The 1-mm milled aliquot was intended for chemical, fermentative, and mycotoxin analysis, while the 0.5-mm milled aliquot was intended for NIR analysis in order to remove spectral noise caused by particle size [72,73].

All 1-mm corn silages sampled were characterized for the presence and quantification of fungal metabolites by using LC-MS/MS at the Department of Agrobiotechnology according to Sulyok et al. [74].

These authors validated a Liquid Chromatography coupled to a tandem Mass Spectrometry (LC-MS/MS) based method to quantify more than 500 secondary microbial metabolites thanks to its high selectivity, sensitivity, robustness, and multi-analyte capability that allow for the determination of a large number of analytes simultaneously.

The analytical procedure consists of weighing 5 g of samples and the subsequent extraction with 20 mL acetonitrile/water/acetic acid (79:20:1, *v/v/v*) for 90 min on a rotary shaker (GFL, Burgwedel, Germany) for 90 min at room temperature in a horizontal position. A volume of 500 µL of the extract was diluted with 500 µL of dilution solvent composed of acetonitrile:water:acetic acid (20:79:1, *v/v/v*) in vials. Then, 5 µL was injected into an LC-MS/MS system with QTrap 5500 MS/MS (Sciex, Foster City, CA, USA) coupled with an Agilent 1290 series UHPLC system (Agilent Technologies, Waldbronn, Germany). Chromatographic separation was performed at 25 °C on a Gemini C18 column (150 × 4.6 mm i.d.,5 μm particle size) equipped with a 4 × 3 mm precolumn with the same characteristics (Phenomenex, Torrance, CA, USA). The eluents used were composed of methanol/water/acetic acid (10:89:1, *v:v:v*) as eluent A and methanol/water/acetic acid (97:2:1, *v:v:v*) as eluent B. Scheduled multiple reaction monitoring was used for analyte detection. Stock solutions of each reference standard of mycotoxin and fungal metabolite were prepared by dissolving the substance, at 250 μg/mL, in acetonitrile with a few compounds dissolved in acetonitrile/water 1:1 (*v/v*), methanol, or water instead. 

Sixty-two intermediate mixes were prepared by mixing the stock solutions of 10 analytes each, but the final multi-analyte standard was freshly prepared prior to spiking experiments by mixing the intermediate mixes. All solutions were stored at −20 °C. External calibration using a serial dilution of a multi-component stock solution was used for the quantification of the mycotoxins. The accuracy of the method was verified by the authors on a continuous basis by participating in ring trials with a current success rate (z-scores between −2 and +2) of approximately 95% for over 1400 results submitted. Moreover, the ion ratio had to agree with the corresponding values of the standards within 30% whereas, for the retention time, a stricter criterion of ±0.03 min was applied in this study.

Samples were also characterized for chemical and fermentative traits, as previously reported by Gallo et al. [20]. The DM was determined by gravimetric loss of free water by heating at 105 °C for 3 h (Association of Official Analytical Chemists or AOAC 1995, method 945.15); ash was determined as a gravimetric residue after incineration at 550 °C for 2 h (AOAC 1995, method 942.05), and ether extract (EE) was obtained following the method 920.29 of AOAC (1995). The crude protein (CP, N × 6.25) was determined using the Kjeldahl method (AOAC 1995, method 984.13). The soluble fraction of CP (expressed on a DM basis) was determined according to Licitra et al. [75]. Neutral detergent fiber (NDF), acid detergent fiber (ADF), and lignin (ADL) were determined using the AnkomII Fiber Analyzer (AnkomTechnology Corporation, Fairport, NY, USA) according to the method described by VanSoest et al. [76]. The NDF analysis utilized a neutral detergent solution containing sodium sulfite and a heat-stable amylase (activity of 17.400 Liquefon units/mL, Ankom Technology, Fairport, NY, USA). NDF, ADF, and ADL contents were corrected for the residual ash content. Starch was measured by polarimetry (Polax 2L, Atago^®^, Tokyo, Japan). The disappearance of NDF after 24 h of rumen incubation (24 h NDFD) was measured in situ by incubating nylon bags in the rumen of two cannulated dairy cows for 24 h [77]. Regarding fermentative parameters, 50 g of wet samples were extracted with a Stomacher blender (Seward Ltd., West Sussex, Worthing, UK) for 3 min in distilled water (water:sample fresh weight ratio: 3:1). After filtering using gauze, an aliquot was centrifuged at 3000 r.p.m. for 15 min at room temperature. A liquid phase added with an internal standard (i.e., pivalic acid) was injected into a gas chromatographic-flame ionization detector (GC/FID) system that was equipped with a capillary column DB-WAX UI (60 m × 0.250 mm; 0.25 µm; Agilent Technologies S.p.A., Milano, Italy). Lactic acid was instead determined using HPLC [78].

A Foss NIRSystem 5000 spectrometer (Foss 5000 NIR systems; Foss Electric, York, UK) was used to collect near-infrared (NIR) spectra. The instrument had a scanning range from 1100 to 2500 nm and diffuse reflectance NIR spectra were obtained with a 2 nm step acquisition (i.e., 700 variables) from 2–3 g of samples ground at 0.5 mm. Ground samples were scanned using a standard ring cup of the FOSS NIR instrument, a small round quartz cup with a 3.75 cm diameter, as a sample holder.

Each sample was scanned twice and then an average of the two acquisitions was used for the present calibration (Appendix A). 

### 5.2. Discriminant Cut-Off Limits Applied to Mycotoxin Groups 

To compare different classification approaches, three different cut-off levels were adopted related to each mycotoxin contamination in order to assign for each sample a high or low level of contamination with respect to these values. With this aim, a high contamination class and a low contamination class were created with samples that had values of mycotoxin infection higher or lower to identified cut-off limits.

The mycotoxin contamination parameters (Table 2) that were used were (i) the total concentration (i.e., μg/kg dry matter or DM) and total count of detectable mycotoxins (i.e., *Aspergillus* toxins, *Alternaria* toxins, zearalenone and its metabolites, trichothecenes type B, fumonisins and their metabolites, enniatins, beauvericin, other emerging Fusarium toxins, penicillium toxins, and other fungi toxins and unspecified fungi toxins); (ii) regulated (i.e., fumonisins B1 and B2, deoxynivalenol, and zearalenone, being those detected in the samples) and emerging Fusarium toxins called R&E- Fusarium toxins count and sum; (iii) only emerging Fusarium toxins called E- Fusarium toxins count and sum; (iv) Fumonisins and their metabolites; and (v) Penicillium toxins count and sum as reported in a previous work by Gallo et al. [20].

In particular, for the total sum of mycotoxin, the cut-off limits were arbitrarily set to 4000, 7000, or 10,000 µg/kg dry matter (DM), and counts were set to 28, 31, or 34. Regulated and emerging *Fusarium*-produced mycotoxins had 1500, 2000, or 2500 µg/kg DM as sum cut-off limits while 13, 14, or 15 were set as count limits. Only emerging *Fusarium*-produced mycotoxins were distinguished with high or low contamination levels with limits of 700, 1000, or 1200 µg/kg DM and 6, 7, or 8. Fumonisin sum and count classes were identified with 400, 700, or 1000 µg/kg DM and 4, 5, or 6 levels, respectively.

Finally, the *Penicillium*-produced mycotoxin classes were identified with contamination levels greater or less than 150, 250, or 350 µg/kg DM and 3, 4, or 5. For each cut-off limit, samples that have mycotoxin concentration values or counts below the threshold are classified as class 1, or if higher than the used threshold, as class 2.

### 5.3. Outlier Spectra Detection, Re-Sampling Procedure, Development, and Evaluation of Classification Models

PCA was used to analyze spectra and detect outliers according to the influence plot, based on Hotelling’s T2 statistic with a 99% confidence interval [79]. This statistical approach is used as a diagnostic tool for out-of-scope sample detection during multivariate model development [80]. The acquired spectra were processed by Unscrambler X version 10.5.1 (CAMO Software AS, Oslo, Norway). 

The thresholds previously specified created an imbalanced class distribution with most of the samples in the lower contamination group, whereas few samples were labeled in the high contamination class, characterized by a higher risk of the presence of mycotoxins. 

To provide comparable size between classes in the data, we applied the Synthetic Minority Over-sampling Technique (SMOTE) algorithm [81] as implemented in the ‘smote_and_undersample’ function from the HyperSMURF v2.0 R package [82]. SMOTE is specifically conceived to handle imbalanced data and re-balance classes. Letting $I^{min}$ be the instances of the minority class and $I^{maj}$, the more abundant class, be the high and low mycotoxins contaminated samples, respectively, SMOTE generates realistic new instances by linear interpolation between randomly chosen pairs of close samples in the minority class. Specifically, for every minority instance $x\in \;I^{min}$ the *k* nearest-neighbor instances $KNN\;\subset \;I^{min}\;$are computed and $x^{\prime }\in KNN$ is randomly drawn. Then, a new synthetic instance $x_{S}$ is computed by linear interpolation between them:(1)$$x_{S}=x+\lambda \left(x^{\prime }-x\right)$$
where λ is a random number between 0 and 1. The process is repeated and results in new “synthetic” data $I_{S}^{min}$. The majority class $I^{maj}$ is then resized by random subsample to obtain $I^{maj\prime }$ where $\left|I^{maj\prime }\right|=\left|I^{min}\right|+\left|I_{S}^{min}\right|$.

A classification model to predict high vs. low contamination using wavelength as predictors was fitted through a random forest (RF) as implemented in the ‘randomForest’ R package Version 4.6-14 [83]. 

Here, we applied an RF to a classification problem (categorical response), rather than a regression problem (continuous response). For classification, RF uses *Gini* impurity (*Gini*) to determine the spit at each node of each decision tree, as in:(2)$$Gini=1-\overset{c}{\underset{i=1}{\sum }}{\left(p_{i}\right)}^{2}$$
where *p_i_* represents the relative frequency of the class observed and *c* represents the number of classes. Each decision tree selected a set of important predictors *f_b_*, which were bagged (bootstrapped and aggregated), and the majority voted to a final model *fi* as in
(3)$$\hat{fi}=\frac{1}{B}\overset{B}{\underset{b=1}{\sum }}f_{b}\left(x^{\prime }\right)$$
where *x*’ is the individual decision tree of *B* trees.

Different classification models (e.g., PCA, partial least squares discriminant analysis, neural network, etc.) were tested in a preliminary phase of the study, with RF considered the most promising one. Here, we reported the results obtained by applying RF for classification. In particular, the class-balanced dataset was split into training and testing subsets including 70% and 30% of the samples, respectively. The RF generated several uncorrelated decision trees, each evaluating a random subset of features. The results of the trees were then averaged to provide the final model. 

To evaluate the accuracy of the RF classification model, sensitivity and specificity were calculated [84]. Accuracy was the ability of the model to differentiate the high and low contamination samples. The sensitivity of the model was its ability to determine the low contamination cases correctly. To estimate it, we calculated the proportion of correctly low classified samples with the total of low classified samples by model. Specificity was the same as sensitivity but regarded the highest class.

Moreover, the interval of confidence (CI) and p-value of discriminant models were reported. The 95% confidence interval (CI) of the overall accuracy rate was provided including it and a one-sided test to assess if the accuracy was better than the “no information rate” which is taken to be the largest class percentage in the data (CARET package in R studio software).

## Figures and Tables

**Figure 1 toxins-14-00323-f001:**
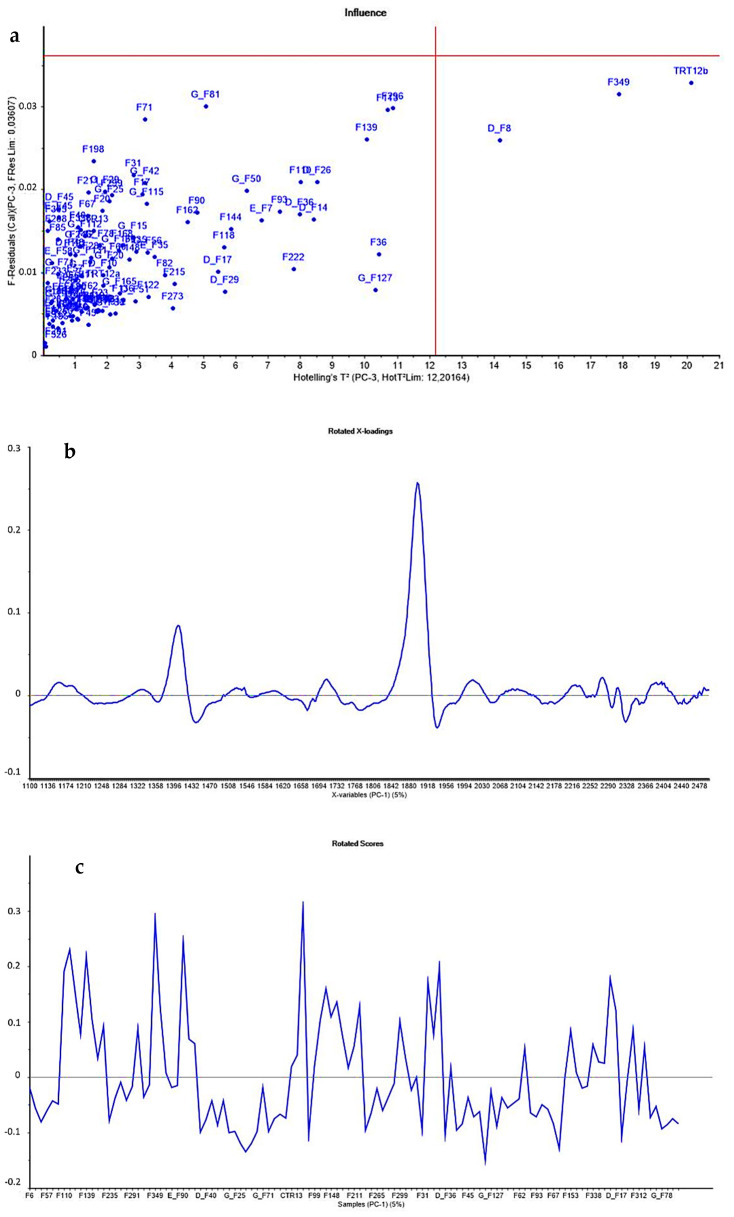
PCA results: (**a**) Hotelling’s T2 values and F-residuals plot were performed in principal component analysis to detect spectra outliers with an interval of confidence of 99%; (**b**) rotated loadings; and (**c**) rotated subject scores.

**Table 1 toxins-14-00323-t001:** Chemical, biological, and fermentative traits (% DM) characterizing corn silages.

	Chemical and Biological Parameters (% DM)	
Items	Mean	Standard Deviation
**DM (% fresh matter)**	34.34	2.38
**Ash**	5.78	0.13
**CP**	8.26	0.44
**EE**	2.94	0.09
**NDF**	37.24	1.27
**ADF**	24.74	0.92
**ADL**	3.00	0.16
**NDIP**	1.02	0.13
**ADIP**	0.74	0.09
**24 h NDFD (% NDF)**	50.64	1.80
**Starch**	31.54	2.77
	**Fermentative and Organoleptic Parameters (% DM)**	
	**Mean**	**Standard Deviation**
**pH (dmnl)**	3.82	0.13
**Acetic acid**	3.19	0.50
**Propionic acid**	0.18	0.14
**Butyric acid**	0.005	0.003
**Lactic acid**	3.21	0.82
**Lactic to Acetic**	1.32	0.59
**Ethanol**	0.52	0.13
**1,2 propanediol**	0.50	0.21
**N-NH3 (% Total N)**	10.46	2.56

**Table 2 toxins-14-00323-t002:** Descriptive statistics for sums (µg/kg DM) and counts (n) of Regulated and Emerging mycotoxins in corn silage in the original database.

Items	Mean	Sd ^3^	Skewness	Kurtosis	25% Percentile	50% Percentile	75% Percentile
**Sum of mycotoxins**	5895.70	7252.46	2.08	4.37	1208.68	2643.24	7235.01
**Sum of R&E-*Fusarium* toxins ^1^**	4781.04	6539.44	2.33	5.54	981.76	2077.46	5446.44
**Sum of E-*Fusarium* toxins ^2^**	2453.83	3571.47	2.82	8.19	641.81	1187.43	2125.63
**Sum of Fumonisins**	2181.59	3430.07	2.25	4.58	256.67	620.21	2476.63
**Sum of *Penicillium* toxins**	177.74	221.88	2.25	6.44	30.66	67.67	243.14
**Count of mycotoxins**	26.20	6.42	0.44	−0.07	21.50	26.00	30.50
**Count of R&E-*Fusarium* toxins ^1^**	15.33	3.61	0.51	1.67	13	16	17
**Count of E-*Fusarium* toxins ^2^**	7.02	1.73	0.00	0.76	6	7	8
**Count of Fumonisins**	5.45	1.61	−0.46	−0.25	4	6	7
**Count of *Penicillium* toxins**	3.61	1.25	0.26	−0.16	3	4	4

^1^ R&E-*Fusarium* toxins, Regulated and Emerging -*Fusarium* toxins: ^2^ E-*Fusarium* toxins, Emerging *Fusarium* toxins ^3^ Sd, Standard Deviation.

**Table 3 toxins-14-00323-t003:** Near-infrared spectroscopy calibration parameters from the prediction set on the total sum (µg/kg DM) and count (n) of Mycotoxin class ^1^.

TOTAL SUM of Mycotoxins						
	**Cut off 4000 µg/kg DM**		**Cut off 7000 µg/kg DM**		**Cut off 10,000 µg/kg DM**	
	**Class1**	**Class2**	**Class1**	**Class2**	**Class1**	**Class2**
**Class1**	25.7 ± 2.7	6.3 ± 2.7	35.6 ± 2.4	5.4 ± 2.4	42.9 ± 2.1	3.1 ± 2.1
**Class2**	4.6 ± 2.3	25.5 ± 2.3	1.3 ± 1.5	35.8 ± 1.5	0.4 ± 0.9	40.7 ± 0.9
**Accuracy**	82.2 ± 5.9%		91.5 ± 3.5%		96.0 ± 2.7%	
**Sensitivity**	80.2 ± 8.5%		86.9 ± 5.9%		93.2 ± 4.6%	
**Specificity**	81.3 ± 8.0%		96.6 ± 4.0%		99.2 ± 2.2%	
**CI ^2^**	(70.4 ± 6.9%), (90.6 ± 4.3%)		(83.1 ± 4.4%), (96.5 ± 2.2%)		(89.7 ± 3.7%), (98.8 ± 1.4%)	
***p* value**	<0.05		<0.05		<0.05	
**TOTAL COUNT of mycotoxins**						
	**Cut off n = 28**		**Cut off n = 31**		**Cut off n = 34**	
	**Class1**	**Class2**	**Class1**	**Class2**	**Class1**	**Class2**
**Class1**	30.9 ± 2.4	5.1 ± 2.4	38.9 ± 1.7	4.2 ± 1.7	46.3 ±1.7	2.7 ±1.7
**Class2**	3.2 ± 2.2	28.9 ± 2.2	0.6 ± 1.1	38.4 ± 1.1	0.03 ± 0.3	44.0 ± 0.3
**Accuracy**	87.8 ± 4.4%		94.2 ± 2.6%		97.1 ± 1.8%	
**Sensitivity**	85.7 ± 6.7%		90.4 ± 3.9%		94.6 ± 3.5%	
**Specificity**	90.2 ± 6.7%		98.5 ± 2.8%		99.9 ± 0.7%	
**CI ^2^**	(77.8 ± 5.3%), (94.3 ± 3.0%)		(86.9 ± 3.4%), (98.1 ± 1.5%)		(91.5 ± 2.7%), (99.3 ± 0.7%)	
***p* value**	<0.05		<0.05		<0.05	

^1^ Class 1, class of samples lower than cut-off limits; Class 2, class of samples higher than cut-off limits ^2^ CI, Confidence Interval 95%.

**Table 4 toxins-14-00323-t004:** Near-infrared spectroscopy calibration parameters from the prediction set on the sum (µg/kg DM) and count (n) of the Regulated and Emerging *Fusarium*-toxins class ^1^.

SUM of R&E-*Fusarium* Toxins ^2^						
	**Cut off 1500 µg/kg DM**		**Cut off 2000 µg/kg DM**		**Cut off 2500 µg/kg DM**	
	**Class1**	**Class2**	**Class1**	**Class2**	**Class1**	**Class2**
**Class1**	17.4 ± 2.4	6.6 ± 2.4	25.2 ± 2.5	6.8 ± 2.5	29.89 ± 2.53	8.11 ± 2.53
**Class2**	5.1 ± 2.7	23.9 ± 2.7	3.7 ± 2.5	35.3 ± 2.5	2.86 ± 2.44	43.14 ± 2.44
**Accuracy**	77.9 ± 6.1%		85.1 ± 4.8%		86.9 ± 4.0%	
**Sensitivity**	72.6 ± 10.0%		78.6 ± 7.8%		78.7 ± 6.7%	
**Specificity**	82.4 ± 9.2%		90.4 ± 6.5%		93.8 ± 5.3%	
**CI ^3^**	(64.6 ± 6.8%), (88.0 ± 4.7%)		(74.8 ± 5.6%), (92.3 ± 3.5%)		(77.9 ± 4.7%), (93.2 ± 2.9%)	
***p* value**	<0.05		<0.05		<0.05	
**Count of R&E-*Fusarium* toxins ^2^**						
	**Cut off n = 13**		**Cut off n = 14**		**Cut off n = 15**	
	**Class1**	**Class2**	**Class1**	**Class2**	**Class1**	**Class2**
**Class1**	12.8 ± 2.0	5.2 ± 2.0	18.5 ± 2.4	7.49 ± 2.35	24.1 ± 2.5	7.9 ± 2.5
**Class2**	5.2 ± 2.7	15.8 ± 2.7	5.3 ± 2.7	25.67 ± 2.73	4.7 ± 2.7	33.3 ± 2.7
**Accuracy**	73.3 ± 7.7%		77.5 ± 5.6%		82.0 ± 4.9%	
**Sensitivity**	71.2 ± 10.9%		71.2 ± 9.1%		75.4 ± 7.8%	
**Specificity**	75.1 ± 12.9%		82.8 ± 8.8%		85.6 ± 7.2%	
**CI ^3^**	(57.0 ± 8.3%), (85.9 ± 5.8%)		(64.6 ± 6.2%), (87.4 ± 4.3%)		(71.1 ± 5.6%), (90.1 ± 3.8%)	
***p* value**	0.049 ± 0.091		<0.05		<0.05	

^1^ Class 1, class of samples lower than cut-off limits; Class 2, class of samples higher than cut-off limits ^2^ R&E-*Fusarium* toxins, Regulated and Emerging -*Fusarium* toxins ^3^ CI, Confidence Interval 95%.

**Table 5 toxins-14-00323-t005:** Near-infrared spectroscopy calibration parameters from the prediction set on the sum (µg/kg DM) and count (n) of Emerging *Fusarium*-toxins class ^1^.

SUM of E-*Fusarium* Toxins ^2^						
	**Cut off 700 µg/kg DM**		**Cut off 1000 µg/kg DM**		**Cut off 1200 µg/kg DM**	
	**Class1**	**Class2**	**Class1**	**Class2**	**Class1**	**Class2**
**Class1**	13.3 ± 2.3	6.67 ± 2.28	23.8 ± 2.7	8.3 ± 2.7	27.1 ± 2.8	8.0 ± 2.8
**Class2**	3.3 ± 1.9	33.66 ± 1.92	2.2 ± 1.8	56.8 ± 1.8	1.5 ± 1.7	63.6± 1.7
**Accuracy**	82.4 ± 5.2%		88.5 ± 3.1%		90.6 ± 3.1%	
**Sensitivity**	66.7 ± 11.3%		74.2 ± 8.5%		77.3 ± 8.0%	
**Specificity**	91.0 ± 5.2%		96.2 ± 3.1%		97.8 ± 2.7%	
**CI ^3^**	(70.2 ± 6.1%), (91.1 ± 3.7%)		(80.1 ± 3.8%), (94.1 ± 2.2%)		(83.2 ± 3.8%), (95.4 ± 2.2%)	
***p* value**	0.019 ± 0.046		<0.05		<0.05	
**COUNT of E-*Fusarium* toxins ^2^**						
	**Cut off n = 6**		**Cut off n = 7**		**Cut off n = 8**	
	**Class1**	**Class2**	**Class1**	**Class2**	**Class1**	**Class2**
**Class1**	16.0 ± 2.2	7.0 ± 2.3	30.6 ± 2.9	8.4 ± 2.9	51.1 ± 2.2	3.9 ±2.2
**Class2**	5.2 ± 2.3	21.8 ± 2.3	2.26 ± 2.18	51.7 ± 2.2	0	76
**Accuracy**	75.7 ± 5.8%		88.5 ± 3.8%		97.0 ± 1.7%	
**Sensitivity**	69.7 ± 9.9%		78.4 ± 7.3%		92.9 ± 4.0%	
**Specificity**	80.9 ± 8.5%		95.8 ± 4.0%		100.0%	
**CI ^3^**	(61.6 ± 6.4%), (86.6 ± 4.5%)		(80.3 ± 4.5%), (94.1 ± 2.8%)		(92.6 ± 2.4%), (99.1 ± 0.9%)	
***p* value**	<0.05		<0.05		<0.05	

^1^ Class 1, class of samples lower than cut-off limits; Class 2, class of samples higher than cut-off limits ^2^ E-*Fusarium* toxins, Emerging *Fusarium* toxins ^3^ CI, Confidence Interval 95%.

**Table 6 toxins-14-00323-t006:** Near-infrared spectroscopy calibration parameters from the prediction set on the sum (µg/kg DM) and count (n) of fumonisins mycotoxin class ^1^.

SUM of Fumonisins						
	**Cut off 400 µg/kg DM**		**Cut off 700 µg/kg DM**		**Cut off 1000 µg/kg DM**	
	**Class1**	**Class2**	**Class1**	**Class2**	**Class1**	**Class2**
**Class1**	13.5 ± 2.1	5.5 ± 2.1	27.1 ± 3.0	7.0 ± 3.0	34.8 ± 2.7	6.2 ± 2.7
**Class2**	5.2 ± 2.2	11.8 ± 2.2	4.9 ± 2.5	26.1 ± 2.5	2.9 ± 2.3	34.1 ± 2.3
**Accuracy**	70.5 ± 7.3%		81.8 ± 5.2%		88.3 ± 4.3%	
**Sensitivity**	70.8 ± 11.1%		79.6 ± 8.7%		84.9 ± 6.6%	
**Specificity**	69.3 ± 12.8%		84.3 ± 8.0%		92.2 ±6.3%	
**CI ^2^**	(52.8 ± 7.6%), (84.0 ± 5.7%)		(70.44 ± 6.0%), (90.1 ± 3.9%)		(79.2 ± 5.2%), (94.3 ± 3.0%)	
***p* value**	0.074 ± 0.121		<0.05		<0.05	
**COUNT of Fumonisins**						
	**Cut off n = 4**		**Cut off n = 5**		**Cut off n = 6**	
	**Class1**	**Class2**	**Class1**	**Class2**	**Class1**	**Class2**
**Class1**	13.5 ± 2.1	4.5 ± 2.1	25.0 ± 2.1	6.0 ± 2.1	45.2 ± 2.9	5.8 ± 2.9
**Class2**	5.6 ± 2.0	9.4 ± 2.0	5.7 ± 2.6	21.3 ± 2.6	1.0 ± 1.4	43.0 ± 1.4
**Accuracy**	69.5 ± 7.8%		79.8 ± 5.3%		92.8 ± 3.3%	
**Sensitivity**	75.2 ± 11.7%		80.6 ± 6.8%		88.7 ± 5.6%	
**Specificity**	62.7 ± 13.5%		78.9 ± 9.6%		97.6 ± 3.1%	
**CI ^2^**	(51.3 ± 8.2%), (84.0 ± 5.9%)		(67.3 ± 5.9%), (89.0 ± 4.0%)		(85.8 ± 4.2%), (96.9 ± 2.2%)	
***p* value**	0.122 ± 0.153		<0.05		<0.05	

^1^ Class 1, class of samples lower than cut-off limits; Class 2, class of samples higher than cut-off limits ^2^ CI, Confidence Interval 95%.

**Table 7 toxins-14-00323-t007:** Near-infrared spectroscopy calibration parameters from the prediction set on the sum (µg/kg DM) and count (n) of *Penicillium* mycotoxin class ^1^.

SUM of *Penicillium* Toxins						
	**Cut off 150 µg/kg DM**		**Cut off 250 µg/kg DM**		**Cut off 350 µg/kg DM**	
	**Class1**	**Class2**	**Class1**	**Class2**	**Class1**	**Class2**
**Class1**	24.9 ± 2.4	7.1 ± 2.4	36.8 ± 2.1	4.2 ± 2.1	42.8 ± 2.3	4.3 ± 2.3
**Class2**	4.5 ± 2.7	24.5 ± 2.7	1.3 ± 1.8	35.7 ± 1.8	0.1 ± 0.7	41.9 ± 0.7
**Accuracy**	81.0 ± 5.8%		92.9 ± 3.4%		95.1 ± 2.8%	
**Sensitivity**	77.9 ± 7.4%		89.8 ± 5.2%		91.0 ± 4.9%	
**Specificity**	84.4 ± 9.2%		96.4 ± 4.8%		99.7 ± 1.6%	
**CI ^2^**	(69.1 ± 6.6%), (89.7 ± 4.4%)		(85.0 ± 4.4%), (97.3 ± 2.0%)		(88.5 ± 3.8%), (98.4 ± 1.5%)	
***p* value**	<0.05		<0.05		<0.05	
**COUNT of *Penicillium* toxins**						
	**Cut off n = 3**		**Cut off n = 4**		**Cut off n = 5**	
	**Class1**	**Class2**	**Class1**	**Class2**	**Class1**	**Class2**
**Class1**	24.0 ± 2.6	6.0 ± 2.6	42.5 ± 2.4	4.5 ± 2.4	56.4 ± 1.5	1.6 ± 1.5
**Class2**	5.2 ± 2.1	21.8 ± 2.1	0.9 ± 1.4	41.4 ± 1.4	0	52.0 ± 0
**Accuracy**	80.2 ± 5.3%		94.0 ± 3.1%		98.6 ± 1.3%	
**Sensitivity**	79.9 ± 8.6%		90.5 ± 5.0%		97.3 ± 2.5%	
**Specificity**	80.6 ±7.8%		97.9 ± 3.3%		100.00%	
**CI ^2^**	(67.7 ± 5.9%), (89.5 ± 4.0%)		(87.0 ± 4.0%), (97.7 ± 1.8%)		(94.3 ± 2.1%), (99.7 ± 0.5%)	
***p* value**	<0.05		<0.05		<0.05	

^1^ Class 1, class of samples lower than cut-off limits; Class 2, class of samples higher than cut-off limits ^2^ CI, Confidence Interval 95%.

**Table 8 toxins-14-00323-t008:** Bibliographic information (i.e., source) published in the last three years regarding the use of NIR for predicting mycotoxin contaminations in different matrices.

Feed Matrix	Target Mycotoxin	Wavelength	Statistical Model *	Results Obtained	Practical Application	Source
**Ground corn samples**	Fumonisin B1 and B2	900–1700 nm	PLS, SVM, and LPLS-S	R^2^ prediction = 0.71–0.91 RMSEP = 12.08–22.58 mg/kg	Pocket-sized NIR spectrometers controlled by a smartphone	[65]
			PCA, PLS-DA, and SVM-DA	Prediction accuracy = 86.3–88.2% Error in prediction = 11.8–13.7%		
**Rice (*Oryza sativa* L.)**	Aflatoxin B1	400–2498 nm	MSA + PLS	Low-aflatoxin-level (≤35 μg/kg): R^2^ calibration = 0.72–0.99 RMSEC = 0.11–5.02 μg/kg High-aflatoxin-level (>35 μg/kg): R^2^ calibration = 0.72–0.99 RMSEC = 0.56–13.74 μg/kg	Monitoring aflatoxin B1 contamination in milled rice during postharvest storage	[66]
**Almonds**	Aflatoxin B1	900–1700 nm	PLS	R^2^ = 0.786–0.958 RMSEP = 0.089–0.197 μg/g	Commercial application	[67]
**Distiller’s dried grains**	Fumonisin B1 and B2	400–2500 nm	PLS	FB1 R^2^ = 0.80 FB2 R^2^ = 0.79	Potential to support decision making regarding the use of feed ingredients and, consequently, to protect animal health	[68]
**Barley (*Hordeum vulgare*)**	Deoxynivalenol (cut off limit cut off 1250 µg/kg)	10,000 cm^−1^–4000 cm^−1^	PLS-DA	Sensitivity in cross-validation = 90.9% Specificity in cross-validation = 89.9%	Green technique to monitor DON contamination	[69]
**Corn products**	*Fusarium verticillioides* and *F. graminearum*	1000–2500 nm	PLS-DA	Accuracy = 99.7%	Monitoring the safety of feed and food supply	[70]
**Wheat flour**	Deoxynivalenol		PLS-DA and PC-LDA	Contamination level ≤ 450 μg kg^−1^ Accuracy (PLS-DA) = 85–87.5% Error (PLS-DA) = 10–15% error; Accuracy (PC-LDA) = 85% Error (PC-LDA) = 10–15% error	Screening method to evaluate DON contamination to support decision making in industries	[71]

* PLS = Partial least squares; SVM = Support vector machine; LPLS-S = local PLS based on global PLS score; PCA = principal component analysis; PLS-DA = partial least squares discriminant analysis; SVM-DA = support vector machine discriminant analysis; MSA = modified simulated annealing; PC-LDA = Principal Component Analysis-Linear Discriminant Analysis.

## Data Availability

The data that support the findings of this study are available from the corresponding author, [AG], upon reasonable request.

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
