# Peer review of "A Preliminary Study to Classify Corn Silage for High or Low Mycotoxin Contamination by Using near Infrared Spectroscopy"

_toxins, 2022, doi:10.3390/toxins14050323_

Round 1
Reviewer 1 Report
First of all, congratulations to the authors for using this technique for the detection of mycotoxins and the fungi that produce them. We are very interested in it being published, but with the information provided by the authors, it could not be reproduced using other samples. Therefore, I am going to make some suggestions for the improvement of the publication.
One of the suggestions, to improve the reading of the article is the structure. In this sense, it is recommended to put the material and methods behind the Introduction.
Within the results section:
Lines 109-110: An initial PCA is made, but only the result of the influence graph is shown, the samples are not identified in the graph, neither the score of the samples nor the factor loading of the variables appear. It is recommended to display the results in a more normal flow: PCA-Supervised Classification Systems (SIMCA, PLS-DA, LDA). It is also suggested that all quadrants be placed in Figure 1.
Line 130: How was the calibration done?
Within the discussion section:
Line 275; The statistical procedure used is a bit confusing, since in the different data treatments it is not known whether it works with the entire spectrum or with its PCA scores. The SMOTE procedure for rescaling variables together with the “Random Forest” classification system can result in an overestimation of the results. It is not known what Package CARET is, it is assumed to be an R package.
Within the materials and methods section:
Line 374: Drying for 48 hours at 60ºC seems insufficient to me, since it is usual at 65ºC for 4 days, since part of the fungi can grow later, although if the chromatographic analysis is done on these samples there would be no problem.
Line 375: Grinding is done with a 0.5 or 1 mm sieve. It would be necessary to clarify if 1mm is used for chromatography and 0.5mm for NIRS analysis.
Lines 380, 384 and 388: The type of mycotoxins analyzed and the analysis method seems a bit confusing: according to what they say, the chromatographic method classifies more than 500 microbial metabolites, but they do not say which of them they work and in the end they group it in the “regulated and emerging” categories.
Line 392: It does not describe what type of capsule it uses. Also, 2-3 grams seems insufficient Line 451: I would like to know with what criteria they choose the training or test samples.
Appendix A: Claims of association between wavelengths and fungal infection are not supported by any scientific evidence in this article
Reviewer 2 Report
The aim of the research was to obtain qualitative models able to descriminate corn silage for concentration mycotoxins. The "Introduction" chapter provides an overview of the current world literature on this subject, but requires some supplements. The discussion is well conducted and comprehensive. Well-chosen references. Before publishing in Toxins, the article requires additions and corrections. The proposed changes are listed below:
General comments:
Please prepare the article in accordance with the instructions for authors.
Please check that tables 2 to 6 are prepared in accordance with the instructions for the authors.
In tables 1-6, the table header must be in bold
In the References section, for a range of pages, use the long "-" from the insert function for all References items
Please add missing chapters: Funding, Acknowledgments, Author Contributions, Data Availability Statement, Conflicts of Interest
Detiled comments
In the "Introduction" section, add information:
What are the types of corn silage (with cob, without cob, only cob, etc.), what varieties of corn are best suited for silage and which for grain
What is the importance and nutritional value of corn silage for other species of ruminant farm animals (sheep, goat) and monogastric animals (horse, pig, poultry - including geese)
What is the number of mycotoxin-contaminated feed groups for farm animals, where are favorable conditions, what are the possibilities of reducing the contamination of feed with mycotoxins, including silage. What is the effect of lactic fermentation bacteria on reducing silage mycotoxin contamination?
L272 [36-38] instead of current form
L286 [42.43], no spaces
L486 1070-1073?
L488 172-180
L530 no pages
L556 no page range
L572 Is the abbreviated name journal correct?

Reviewer 3 Report
- The traits of the pre-ensiling and post-ensiling datasets should be reported
- What are the NIR units used
- The concentration of mycotoxins and chemical composition of corn silage should include in the results and discussion section
- Authors should include the organoleptic characteristics of corn silage
- Should explain about Regression algorithms
- Calibration and validations of data from NIR spectra
- Comparative studies of previously reported data with current work should mention in a table
Reviewer 4 Report
The paper is of practical interest proposing the validation of a rapid method to detect mycotoxins content in corn samples using NIR method.
However, some aspects can be improved:
1. Please, mention in methodology all LC/MS/MS characteristics: separation, detection, ion of interest, standards including all analysed mycotoxins and so on (Lines 383-388).
2. The discussion should include a comparison between the values ​​obtained by LC/MS/MS and NIR methods.
3. The conclusions should be inserted at the end, after methodology.
Round 2
Reviewer 3 Report
Table 8. Biblioghraphy information (i.e., source) pucclished in the last 3 years that regarded the use of NIR for predicting mycotoxin contaminations in different matrix.
It should be corrected
Reviewer 4 Report
The authors revised the paper according the requirements. I recommend its publication in this form.